# When Natural Compounds Meet Nanotechnology: Nature-Inspired Nanomedicines for Cancer Immunotherapy

**DOI:** 10.3390/pharmaceutics14081589

**Published:** 2022-07-30

**Authors:** Linna Yu, Yi Jin, Mingjie Song, Yu Zhao, Huaqing Zhang

**Affiliations:** 1People’s Hospital of Qianxinan Buyi and Miao Minority Autonomous Prefecture, Xingyi 562400, China; linna.y@stu.cpu.edu.cn; 2Key Laboratory of Drug Quality Control and Pharmacovigilance (Ministry of Education), State Key Laboratory of Natural Medicines, Department of Pharmaceutics, NMPA Key Laboratory for Research and Evaluation of Pharmaceutical Preparations and Excipients, China Pharmaceutical University, Nanjing 210009, China; cpu_jinyi@stu.cpu.edu.cn (Y.J.); songmj98@stu.cpu.edu.cn (M.S.)

**Keywords:** natural compounds, drug delivery, controlled drug release, green materials, cancer immunotherapy

## Abstract

Recent significant strides of natural compounds in immunomodulation have highlighted their great potential against cancer. Despite many attempts being made for cancer immunotherapy, the biomedical application of natural compounds encounters a bottleneck because of their unclear mechanisms, low solubility and bioavailability, and limited efficacy. Herein, we summarize the immune regulatory mechanisms of different natural compounds at each step of the cancer-immunity cycle and highlight their anti-tumor potential and current limitations. We then propose and present various drug delivery strategies based on nanotechnology, including traditional nanoparticles (NPs)-based delivery strategies (lipid-based NPs, micelles, and polysaccharide/peptide/protein-based NPs) and novel delivery strategies (cell-derived NPs and carrier-free NPs), thus providing solutions to break through existing bottlenecks. Furthermore, representative applications of nature-inspired nanomedicines are also emphasized in detail with the advantages and disadvantages discussed. Finally, the challenges and prospects of natural compounds for cancer immunotherapy are provided, hopefully, to facilitate their far-reaching development toward clinical translation.

## 1. Introduction

Existing in nature as a valuable collection of drugs for treating disease, natural compounds share a wide range of pharmacological activities and few side effects with a broad range of safe administration options, individual prescription flexibility, and diversity in preparations [1]. Due to these excellent qualities, natural products have been identified as lead compounds for drug development and direct sources of natural medicines [2]. With a long history of application from the beginning of civilization, the clinical application of natural compounds has formed a global ethnic diversity medical theory system [3,4,5,6,7,8]. In the past 50 years, several typical natural compounds (e.g., ginsenoside Rg3, indirubin, artemisinin, and paclitaxel) have been proven to be highly active at the clinical level [9]. More importantly, more than 65% of anti-tumor drugs are natural compounds [10], such as doxorubicin (DOX), paclitaxel (PTX), and hydroxycamptothecine (HCPT), which have made an indelible contribution to the lives of countless cancer patients.

As one of the four major therapeutic strategies for cancer (along with chemotherapy, surgery, and radiotherapy), cancer immunotherapy has been described as the most promising strategy for conquering cancer by orchestrating the immune system to eradicate cancer cells, including primary tumors, metastasizing tumor, and recurrent tumor. Since the first monoclonal antibody ipilimumab was approved in 2011, cancer immunotherapy has flourished. Compared with other treatment strategies of chemotherapy, surgery, and radiotherapy that are limited by low efficiency, serious side effects, and tumor recurrence, cancer immunotherapy has benefits in orchestrating the body’s immune system to attack tumor cells with improved efficacy, reduced toxicity, and limited side effects. More importantly, by domesticating the immune system, the body is able to gain immune memory that can re-recognize and re-attack the tumor cells in tumor recurrence, leading to a good prognosis [11]. Drug developments for cancer immunotherapy focus on developing not only technology to directly attack tumor cells with effector cells or molecules (such as chimeric antigen receptor (CAR)-T or T cell receptor (TCR)-T) but also cargoes that activate the immune system by targeting different steps for producing an immune response (such as programmed cell death protein 1 (PD1) monoclonal antibodies, cytotoxic T lymphocyte-associated antigen-4 (CTLA4) monoclonal antibodies, cancer vaccines, and cytokines) [12,13,14]. Despite these remarkable advances, current cancer immunotherapy is often limited by safety profiles that cause autoimmunity or nonspecific inflammation in most patients, resulting in large physical trauma and normal tissue destruction [15]. For example, a high dosage of interleukin-2 (IL-2) administration often causes cytokine release syndrome and vascular leak syndrome, leading to serious adverse effects (e.g., hypotension, fever, renal dysfunction, and other potentially lethal actions) with limited clinical tolerability and applications [16]. Therefore, more candidate compounds with specificity and better safety profiles need to be explored.

Attracted by the special immunomodulatory potential of natural compounds, researchers have focused on the application of natural compounds in cancer immunotherapy, trying to build a new landscape of immunotherapy for cancer. In the different steps for generating an anti-cancer immune response, natural compounds play various regulatory roles, including promoting the release of tumor-derived antigen, enhancing the activation and proliferation of immune cells, promoting antigen presentation, etc. [17]. Although natural compounds share merits in extensive pharmacological activities, most of the natural compounds are hydrophobic and unstable because of their polycyclic, polyconjugated structure, or special functional regiments. The application of natural compounds in cancer immunotherapy is severely limited by poor drug solubility, poor membrane permeability, low bioavailability, poor stability, and limited distribution in target sites [18]. For a desired therapeutic effect, a high dose and repeated administration are often required, which often results in poor patient compliance.

To overcome the limitations, drug delivery systems (such as liposomes, nanoparticles, and micelles) have attracted increasing attention in applying natural compounds to anti-tumor immunotherapy because of their superior stability, small size, adjustable surface charge, and functional modification that can improve the solubility, stability, and target capability of natural compounds [19,20]. In addition, natural compounds are often used in combination with other drugs for cancer immunotherapy, which requires concurrently delivering different drugs into target lesions that can be achieved by a drug delivery system [21,22]. Drug delivery systems can transport cargos deeply into a solid tumor, protect drugs from degradation, and achieve on-demand drug release profiles for amplified anti-tumor immunotherapy [23]. Especially for natural compounds that need repeated administration with high doses, the multifunctional drug delivery system can improve the solubility and stability and enhance the targeting and selectivity of natural compounds for improving pharmacological activity and reducing the dosage and frequency of administration [24,25].

In this review, focusing on the immunoregulatory mechanisms, we provide an overview of the significant progress of natural compounds in cancer immunotherapy achieved by versatile nanomaterials (Figure 1). By examining the immunoregulatory mechanisms of natural compounds at each stage of generating a robust anti-cancer immune response, we can understand how to amplify the immunoregulatory activity by exploration of delivery platforms. Representative applications of natural compounds based on nanomedicines are also presented in detail with the advantages and disadvantages discussed. In addition, the possibility of employing emerging delivery strategies (such as carrier-free drug delivery systems) in developing natural compounds inspired by nanomedicine for cancer immunotherapy has also been provided and discussed. This review aims to highlight the fundamental role of natural compounds in cancer immunotherapy and facilitate the transformation of natural products into novel cancer immunomodulators with high efficacy and safety with the help of versatile nanomaterials.

## 2. Immunoregulatory Mechanisms of Natural Compounds in the Cancer-Immunity Cycle

For a robust anticancer immune response, the body goes through a series of progressive events, which are summarized as the “cancer-immunity cycle”. As shown in Figure 2, after generating tumor-derived antigens, antigen-presenting cells (APCs) such as dendritic cells (DCs) capture the antigens and then transport them to draining lymph nodes for activating T cells, which then traffic to tumor sites and specifically recognize cancer cells via the interactions of T cell receptors (TCRs) and major histocompatibility complex class (MHC) molecules, leading to the killing of cancer cells with antigenic release and thus triggering the autoloop of the cancer immunity cycle with a continuous anti-tumor immune effect. However, the cancer-immunity cycle is unsustainable in many cases with one or more of these steps being blocked because of the complexity and diversity of the tumor immune microenvironment (TIME). There is growing evidence that various natural compounds derived from natural products have emerged as potent weapons in the fight against cancer. Accordingly, we will summarize and discuss the immunoregulatory mechanisms of natural compounds in terms of how to initiate or reinstate the cancer-immunity cycle for enhanced cancer immunotherapy.

### 2.1. Natural Compounds Triggered Antigen Release

The starting link in the cancer-immunity cycle is the generation and release of tumor antigens such as tumor-associated antigens (TAAs) and tumor-specific antigens (TSAs). Various natural compounds have demonstrated promising potential in the generation of TAAs and in situ whole-cell “vaccines” with multiple specific epitopes by inducing immunogenic cell death (ICD). The process of ICD is characterized by damage-associated molecular patterns (DAMPs) with the release of several “eat me” signals such as calreticulin (CRT), ultimately initiating an antigen-specific immune response against tumors [26]. ICD inducers also trigger an explosive release of immunostimulatory factors (e.g., adenosine triphosphate (ATP), IL-1β, chemokine (C-X-C motif) ligand 10 (CXCL10), and high-mobility group protein B1 (HMGB1)) to further enhance the immune response.

As a natural anthracycline derived from *actinomycetes*, DOX is the first-line chemotherapy drug used in clinical settings with a broad anti-tumor spectrum. In addition, DOX induces immunogenic apoptosis of cancer cells accompanied by cell surface exposure of CRT and release of heat shock proteins (HSPs) (such as HSP70 and HSP90), as well as secretion of ATP and HMGB1 [27,28], acting as an effective ICD inducer. Nevertheless, the effect of DOX-induced ICD is closely related to the activity of DCs and CD8+ T cells. The immunogenic effect of DOX would be repealed once these immune cells were inhibited. Thus, attention should be paid to improve the selectivity of this cytotoxic compound and avoid its killing effect on the immune cells. Similar to DOX, the natural compound PTX functions as a microtubule-destabilizing agent against several solid tumors and induces ICD-associated DAMP via toll-like receptor 4 (TLR4)/inhibitor of nuclear factor kappa-B kinase 2 (IKK2)/soluble N-ethylmaleimide-sensitive factor attachment protein receptor (SNARE)-dependent exocytosis [29]. Moreover, as a main naphthoquinone active component extracted from the root of *Lithospermum erythrorhizon*, shikonin (SHK) can induce ICD through directly binding with heterogeneous nuclear ribonucleoprotein A1 (hnRNPA1) and inhibiting the post-transcriptional process, thus triggering antigen release [30]. SHK can also induce necroptosis through receptor-interacting serine/threonine-protein kinase (RIPK) pathways and enhance autophagy to up-regulate DAMPs [31]. With the development of the research, more natural compounds have been proven to be ICD inducers. For example, as an intermediary in the photodynamic therapy (PDT), hypericin induces cancer cell death and triggers antigen release through a pathway that differs from chemotherapeutics, which was accompanied by endoplasmic reticulum (ER) stress, high-level reactive oxygen species (ROS), and release/presentation of DAMPs [32,33]. In addition to the above-mentioned anthracyclines, alkaloids, and naphthoquinone, cardiac glycosides (CGs) are also good ICD inducers [34,35,36]. For example, oleandrin, a glycoside drug derived from the leaves of *Nerium oleander*, can trigger ER stress through the PERK/elF2α/ATF4/CHOP pathway and promote DAMPs, leading to the occurrence of anti-tumor immune responses [37].

Despite the excellent ICD-inducing activity of natural compounds, their ICD-associated efficacy is still undermined because of their low solubility, low bioavailability, unavoidable rapid clearance, and unsatisfactory tumor penetration. In addition, several standard chemotherapies such as DOX generally lead to multi-drug resistance [38], which is associated with drug efflux from targeted cancer cells and thus partially suppresses their ICD effect. Therefore, co-treatment with efflux pump inhibitors might be a strategy to reduce or prevent resistance [39]. Most importantly, though single compounds can achieve cancer cell death and antigen release, their ICD induction is not strong enough to drive and maintain the robust immune response. Hence, the combination of ICD-inducing natural compounds with other agents is encouraged to repair and amplify the anti-tumor immune response. For example, chemoimmunotherapy that combined DOX with PD-1 antibodies (aPD-1) also showed synergistic improvement in the anti-tumor effect and provided long-anticipated gains in survival. Furthermore, this strategy reduced the effective concentration of DOX and thereby reduced myelosuppression and cardiotoxicity related to DOX [40]. More importantly, the optimal drug combination and multidrug co-delivery strategy are yet to be explored.

### 2.2. Natural Compounds Promoted Antigen Presentation by APCs

Upon exposure to the released antigens, APCs (e.g., macrophages and DCs) recognize and capture these antigens, triggering their transformation from a resting state to an active/mature state accompanied by a series of phenotypic and functional changes. Meanwhile, antigens taken up by APCs are hydrolyzed into peptides by proteases and transported into the ER, which is then loaded onto the MHC I or II molecules and shuttled to the cell surface [41]. This process is called antigen processing and presentation dominated by APCs, in which DCs are a heterogeneous group of the specialized APCs with the strongest antigen-presenting ability [42]. The antigen presentation is a transitional but indispensable step to initiate and modulate immune responses. However, the tumor microenvironment (TME) affects the maturation of DCs and hampers their antigen processing and presentation, resulting in DC dysfunction and the formation of tumor-associated DCs that lose antigen presentation capability and even facilitate tumor progression [43]. Hence, the strategy that promotes DC maturation and antigen presentation might be a promising method for boosting the anti-tumor immune response. Recently, a variety of natural compounds have emerged as regulators that can promote DC maturation and antigen presentation for cancer immunotherapy.

As a class of macromolecules purified from plants, fungi, and bacteria, polysaccharides have been demonstrated to activate DCs for antigen presentation. For example, polysaccharides extracted from *Ganoderma lucidum* (GLPS) can activate DCs through nuclear factor kappa-B (NF-κB) and mitogen-activated protein kinases (MAPK) pathways with an elevated level of proinflammatory chemokines (e.g., CC chemokine ligand (CCL) 5, CCL20, and CCL19) and cytokines (e.g., IL-12, IL-27, and IL-23A) [44]. As a polysaccharide extracted from mushrooms, polysaccharide Krestin (PSK) possesses immunomodulatory functions and has been approved for cancer treatment in Japan since 1997. PSK can also promote DC maturation with elevated expression of MHC II, CD40, CD80, CD86, CD83, and tumor necrosis factors (TNF) [45]. Recently, a Chinese materia medica preparation, Astragulus injection (AGI), has shown an enhanced anti-tumor effect in a tumor-bearing mouse model when combined with a DC-based vaccine, of which the pharmacological activity is related to the polysaccharides, the major active component in the root of *Astragalus membranaceus*. *Lycium barbarum* is also a medicinal plant with a long history of over 2000 years. Its major active component is *Lycium barbarum* polysaccharides (LBP), which can increase the number of tumor-infiltrating DCs and activate them with enhanced antigen presentation, thus triggering a significant immune response [46]. As a sulfated polysaccharide in *Phaeophyceae*, fucoidan could act as an immunologic adjuvant to promote DC maturation. However, fucoidan alone is not sufficient to boost a robust immune response. Hence, fucoidan acts as an immunologic adjuvant for combination immunotherapy with other immunomodulators. For example, when co-administrated with ovalbumin (OVA) antigen, fucoidan could induce a robust OVA-specific immunologic response to cancer immunotherapy.

These studies reveal the potential of natural compounds in DC-based therapeutic strategies for cancer immunotherapy through boosting antigen processing and presentation. It is worth noting that DCs represent a heterogeneous population of cells. Hence, it is crucial to act precisely on a defined subset of DCs. Most of these compounds function as accelerants of DC maturation. However, DC maturation alone is not sufficient to treat tumors because of the limited number of DCs and TME-related factors (e.g., insufficient tumor antigens, alteration of MHC molecules, and production of immunosuppressive cytokines), which has been validated in clinical failure [47]. Hence, a combination of DC-targeting therapy with additional strategies involving different mechanisms might improve their anti-tumor efficacy. For example, the administration of Flt3L that can induce in vivo expansion of DCs is an alternative approach to be coupled with nature-derived accelerants of DC maturation and ultimately improve subsequent T cell priming [48]. Furthermore, the optimal dosages and compatibility of multidrug combinations have to be explored.

### 2.3. Natural Compounds Promoted T Cell Priming and Activation

The specific peptide-MHC molecule complex presented on APCs is recognized by CD8+ or CD4+ T cells through their interaction with TCRs, respectively, triggering T cell priming and activation. Once activated, CD4+ T cells (also called T helper (Th) cells) differentiate into heterogeneous subsets (e.g., Th1, Th2, regulatory T (Treg) cells, and follicular helper T (Tfh) cells), while CD8+ T cells (also called cytotoxic T lymphocytes (CTLs)) can kill cancer cells through secreting cytokines (e.g., TNF-α and INF-γ) or direct action [49]. In addition to the APC-originating antigen stimulation, costimulatory molecular signals and cytokine support are also required in this process [50]. For example, CD28 can mediate the activation of T cells through binding to the CD80 and CD86 of APCs. However, CTLA4 on T cells with higher affinity with CD80 or CD86 outcompetes CD28 and inhibits the proliferation of T cells. An anti-CTLA4 antibody (e.g., ipilimumab) has been approved by the US Food and Drug Administration (FDA) to promote T cell activation [51]. In addition, other signaling pathways induced by inducible T-cell co-stimulators (ICOS), 4-1BB, OX40, CD27, or glucocorticoid-induced TNF receptor (GITR) also facilitate expansion and cytokine production of T cells [52]. Recently, agonists including nature-derived compounds that target these molecules have been designed to boost T cell priming and activation against tumors, which are evolving as a novel strategy to improve the anti-tumor effect of T cells.

For instance, curcumin (Cur), a natural flavonoid extracted from the plant *Curcuma longa*, has been explored as a modulator of immune systems against tumors by down-regulating the expression of CTLA4 for the inhibition of T cell proliferation [53]. Furthermore, Cur was found to effectively convert Tregs to Th1 cells via repressing the Foxp3 gene transcription and enhancing IFN-γ production in patients, suggesting their regulatory effect on T cells [54,55]. Moreover, Cur facilitated the expansion of central memory T cells and effector memory T cells in tumor-bearing hosts [56]. Though boosting effects of Cur on T cell priming and activation have been reported, its applications are still limited by poor water solubility, instability, and low bioavailability, which are yet to be solved.

Similarly, saponins are natural glycosides that are distributed in high terrestrial plants and possess various pharmacologic properties including immunomodulatory activities. Quil A (QA, an enriched mixture of saponins extracted from *Quillaja Saponaria* Molina) and its purified saponin (quillaja saponin-21 (QS-21)) have been used as adjuvants to stimulate the production of CTLs and Th1 cytokines (IL-2 and IFN-γ) and have been evaluated in clinical trials [57]. QA-based ISCOMATRIX vaccines could increase CD4+ and CD8+ T cells and elicit an anti-tumor effect [58]. However, the clinical application of quillaja saponins is severely limited by instability in water and high toxicity [59].

As an active component existing in numerous plants (e.g., grapes and mulberries), resveratrol (RVT) could induce a decrease in Tregs and downregulated FoxP3+ expression, and increase IFN-γ production [60]. Furthermore, RVT treatment could reduce the proportion of Tregs and switch the production of Th2 cytokines (IL-6 and IL-10) to Th1 cytokines (such as IFN-γ) [61]. However, the applications of resveratrol are still hampered by the poor water solubility and low bioavailability.

Other natural compounds such as artemisinin derived from *Artemisia annua*, catechins extracted from green tea, and triptolide extracted from *Tripterygium wilfordii* Hook also possess antineoplastic activity through increasing CD4+/CD8+ T cells that are primed for subsequent anti-tumor action or inhibiting the expansion and function of Tregs [62,63,64]. Nevertheless, there are still some challenges to be managed. For example, the lack of neo-antigen release or the reduced activity of APCs at the early stage, insufficient infiltration of T cells in the later stages, and the immunosuppressive TME all impede their applications. In this respect, the combination of T cell agonist-based treatment with other therapies might be a promising option [50], of which the sequence and dosage of compatible drugs need to be further optimized.

### 2.4. Natural Compounds Promoted T Cell Trafficking and Infiltration into Tumors

Once T cells have been primed and activated, they must traffic and infiltrate into tumors by a series of biochemical clues to beat cancer cells. Recently, “cold tumors” that are characterized by few or no CTLs in tumor sites have been identified, which are further classified into immune-excluded tumors (T cells migrate to the periphery of tumors but fail to infiltrate) and immune-desert tumors (entirely devoid of T cell infiltration) [65]. The limiting factors involved in this process include mismatching of chemokine-chemokine receptor pairs, aberrant vasculature, dense stroma, and immunosuppressive factors in the TME [66]. Recently, several strategies that can convert “cold tumors” into “hot tumors” with increased CLTs in the tumor bed through promoting T cell trafficking and infiltration have been explored vigorously, including chemokines and antiangiogenesis agents [67,68]. For example, anti-vascular endothelial growth factor (VEGF) agents (e.g., apatinib and bevacizumab) have shown positive effects on tumor vascular normalization and rescued infiltration of T cells into the tumor bed with an increase in chemokines (e.g., CXCL1) [69,70]. However, the efficacy comes with some safety issues such as hypertension and hemorrhage. Fortunately, some natural compounds with unique pharmacological activities and low toxicity have emerged as alternatives to improve T cell trafficking and infiltration.

Restoring impaired chemotaxis of activated immune cells facilitates their accumulation at the tumor sites. For example, the glycoprotein derived from an *Azadirachta indica* (neem) leaf preparation (NLGP) is a potent inducer of IFN-γ that can regulate the CXC chemokine family such as CXCL10. NLGP treatment could maintain the physiological homeostasis of CXCL10 with increased secretion of IFN-γ and up-regulate CXCR3A to further enhance immune cell migration [71]. Another promising herbal medicine, *Epimedium*, possesses various pharmacological activities such as anti-tumor and immunomodulatory effects. Huang et al. analyzed 16 active ingredients purified from *Epimedium* using system pharmacology models and established that icaritin (ICT) could promote infiltration of CD8+ T cells into TME by up-regulating chemotaxis (e.g., CXCL9 and CXCL 10), resulting in inhibition of cancer in Lewis lung cancer (LLC) tumor-bearing mice with downregulated immunosuppressive cytokines (e.g., TNF-α, IL-10, and IL-6) [72]. However, its poor solubility, unsatisfactory oral absorption, and low bioavailability are still major limitations for further utilization. In addition, some research indicates that antibiotics (e.g., bleomycin, erythromycin, clarithromycin, and azithromycin) extracted from microorganisms can up-regulate the expression of adhesion molecules (e.g., intercellular cell adhesion molecule-1, vascular cell adhesion molecule, and E-selectin) in endothelial cells that mediate immune cell adhesion and recruitment [73,74].

Another urgent issue is the insufficient infiltration of T cells, especially for immune-excluded tumors. Aberrant vasculature sometimes leads to T cell exclusion. In this respect, vascular normalization by balancing pro- and antiangiogenic factors has emerged as a promising strategy to facilitate CTL infiltration [75]. For example, RVT could function as a natural anti-VEGF agent to decrease the level of VEGF and inhibit angiogenesis in TME at low doses for the accumulation of CD8+ T cells in tumor beds in vivo [61]. Another potential target for antiangiogenesis is hypoxia-inducible factor-1 alpha (HIF-1α). Several natural compounds, including flavonoids, quercetin, skullcap, and Cur could inhibit VEGF production and angiogenesis through down-regulating HIF-1α-mediated signal pathways [76,77,78]. However, consideration should be given to how to achieve targeted delivery of these compounds to the tumor site to avoid common adverse reactions of vasoinhibitors. It is worth noting that the direct impact of the involved mechanisms of antiangiogenesis agents on T cell infiltration needs further clarification. The validation of their applicability to different solid tumors is also necessary.

### 2.5. Natural Compounds Promoted Recognition of Cancer Cells

After infiltration of effector T cells into the tumor bed, CTLs recognize cancer cells via the interaction between TCRs in T cells and the specific peptide-MHC-I molecule complexes in cancer cells. This step is the prerequisite for triggering the CTL-mediated elimination of tumor cells. In recent years, novel therapies such as adoptive cellular therapy based on chimeric antigen receptors (CAR) and T cell receptors (TCR) have prompted great interest, in which the recognition of cancer cells was enhanced by operating the recognition modules on the surface of T cells. However, the variable (neo)antigens and decreased antigens in the cancer cells that are due to impaired antigen presentation (e.g., defects in β-2-microglobulin, tapasin, and transporter associated with antigen processing) often limit the clinical efficacy [79]. Therefore, a combination of adoptive cells with adjuvants that enhance the antigen presentation in the targeted cells might promote the recognition of cancer cells by T cells. Some natural compounds offer alternative candidates to this strategy.

For example, bryostatin 1, a macrocyclic lactone extracted from the marine bryozoan *Bugula neritina*, is a protein kinase C (PKC) modulator that promotes the expression of surface antigens in several cell types and thus enhanced immunogenicity and recognition of cancer cells in chronic lymphoblastic leukemia, acute lymphoblastic leukemia, and non-Hodgkin’s lymphoma [80]. Hardman et al. synthesized a series of analogs based on bryostatin 1 using a function-oriented synthesis approach and found that the selected analogs could elevate the expression level of CD22 (the potent target for CARs in leukemias), providing a novel strategy for CARs-based immunotherapy with enhanced recognition of cancer cells [81]. Another natural compound, atractylenolide I (ATT-I), extracted from *Atractylodes macrocephala* Koidz, can promote antigen presentation on colorectal cancer (CLC) cells. ATT-I binds with the proteasome 26S subunit non-ATPase 4 (PSMD4) to augment the immunoproteasome activity for antigen processing and MHC-I-mediated antigen presentation, leading to enhanced recognition by T cells [82]. It is worth noting that the exploration of natural compounds for augmented recognition of cancer cells is only beginning with a few available compounds. Therefore, more natural compounds and their derivatives with relevant potential and the involved mechanisms remain to be explored.

### 2.6. Natural Compounds Promoted the Killing of Cancer Cells

Tumor-infiltrating CTLs and natural killer cells (NKs) are the major mediators participating in killing cancer cells through granule exocytosis (granzymes and perforin) and death ligand/death receptor systems (Fas ligand or TNF-related apoptosis-inducing ligand (TRAIL)) [83]. However, many types of cancer cells escape immunosurveillance by exploiting immune checkpoint pathways (e.g., PD-1/PD-L1 axis), leading to an impaired ability of immune cells to kill the cancer cells. Though some immune checkpoint inhibitors (ICI) such as monoclonal antibodies (mAbs) targeting PD-1 (e.g., cemiplimab, nivolumab, and pembrolizumab) or PD-L1 (e.g., atezolizumab, avelumab, and durvalumab) have been approved by the US Food and Drug Administration (FDA) and achieved positive results in clinical practices; disadvantages such as severe immune-related adverse events (e.g., myocarditis, liver damage, and neurotoxicity), poor tumor permeability because of high molecular weight (~150 kD), and complex manufacture processes or high production costs have restricted their clinical application [84,85]. Alternatively, some natural compounds with pharmacological activities that inhibit immune checkpoints have been explored as alternatives and are receiving increased attention because of their easy availability and high security.

Apigenin, a flavonoid derived from onions, oranges, or chamomile tea, can attenuate IFN-γ- or IFN-β-induced PD-L1 upregulation in cancer cells by blocking the signal transducer and activator of the transcription 1 (STAT1) pathway, resulting in enhanced T-cell-mediated cancer cell death [86]. Anti-PD-1 mAb combined with apigenin could effectively enhance the cytotoxic effect of CD8+ T cells with an increased level of TNF-α, IFN-γ, and granzyme B in the Lewis lung carcinoma model that did not respond to treatment with anti-PD-1 mAb alone [87]. Berberine (BBR), an isoquinoline alkaloid derived from *Coptidis chinensis* Franch and *Phellodendron chinense* Schneid, exerts an anti-tumor effect on many cell types (e.g., non-small-cell lung cancer (NSCLC) cells and breast cancer cells), of which the anti-tumor effect is abolished in T-cell-deficient mice models, suggesting that immunomodulatory effects of BBR are associated with T cell action [88]. Liu et al. found that BBR could promote PD-L1 degradation through the ubiquitin/proteasome-dependent pathway and inhibit the PD-1/PD-L1 axis through deubiquitination activity, resulting in effective down-regulation of PD-L1 expression and counteraction of IFN-γ-induced PD-L1 upregulation in vitro and in vivo [89]. Pigallocatechin gallate (EGCG), a polyphenol derived from *Camellia sinensis* (green tea) (Figure 3A), has been shown to have an inhibitory effect on PD-L1 of NSCLC cells through the Janus kinase 2 (JAK2)/STAT1 pathway and AKT pathway (Figure 3B–D) [90]. The EGCG treatment restored the tumoricidal function of CD8+ T cells by inhibiting the PD-1/PD-L1 axis in NSCLC-bearing mice (Figure 3E,F) [91]. Moreover, natural compounds RVT can hinder PD-L1 localization at the plasma membrane and promote PD-L1 digestion by disrupting N-glycan branches, thus blocking the PD-1/PD-L1 pathway and promoting T-cell-mediated tumoricidal action [92]. Other natural compounds such as silibinin, Gaertn, and triptolide also demonstrate an immunoregulatory effect on T cells or cancer cells by immune checkpoint inhibition against cancer, suggesting that natural compounds might be a potential alternative to resolve the issues in immune checkpoint blockade (ICB) therapy [93]. Currently, most natural compounds are used as complementary agents in ICB treatment, but there is still a lack of comprehensive understanding of their pharmacological property and action mechanisms. Furthermore, the problem of how to optimize synergies and ensure safety needs to be considered in the future.

Some cancer patients do not respond to immune checkpoint blocking therapy, suggesting that there may be other tumor immunosuppression and tolerance pathways. Indoleamine 2,3-dioxygenase (IDO), which is overexpressed in tumors (e.g., pancreatic cancers and NSCLC) and associated with poor prognosis in cancer patients, has been identified as a motivator for cancer immunosuppression via suppressing local CLTs and NKs, thus making IDO a potential target [94]. Some IDO inhibitors (e.g., epacadostat, navoximod, and indoximods) have entered clinical trials and achieved promising initial results, although common adverse effects such as vomiting, abdominal pain, and dyspnea have also been observed, promoting the exploitation of novel agents such as natural compounds [95]. For example, EGCG can inhibit IFN-γ-induced IDO expression in human colorectal cancer (CRC) by suppressing STAT1 phosphorylation [96]. Brassing, a phytoalexin isolated from Chinese cabbage inoculated with *Pseudomonas chichorii*, has been identified as a natural IDO inhibitor for active host T cell immunity. Banerjee et al. reported that compared to brassing alone, which did not influence the growth of the B16-F10 tumor, treatment of its derivative (5-bromo-brassinin) could induce tumor extinction via active host T cell immunity by inhibiting IDO [97], providing a new suggestion of using natural products as lead compounds for drug exploration. However, IDO is not the only immune whistle in vivo and the anti-tumor effect of IDO inhibitors alone is often limited because of alternative mechanisms bypassing IDO function. In this respect, the combination of IDO inhibitors and other vaccines or ICI (such as PD-1 or PD-L1 inhibitors) will be a trend of natural IDO-inhibitor development [98].

Taken together, natural compounds possess excellent regulatory activity in both innate and adaptive immune systems, participating in one or more steps of the cancer-immunity cycle. Nonetheless, most natural medicines often encounter obstacles, such as poor solubility and stability, low bioavailability and bioactivity, and limited tumor-targeting performance, which limit the clinical translation of natural products and raise demands for better drug performance and tumor-targeted delivery.

## 3. Nature-Inspired Nanomedicine for Cancer Immunotherapy

Multifunctional drug delivery systems (e.g., liposomes, micelles, and biomolecule-based nanoparticles) with excellent performance (e.g., composition, particle size, charge, and various modification sites) have been explored for improving the pharmaceutical properties of natural compounds, facilitating tumor-targeted drug delivery, penetrating biological barriers, and achieving combinatorial therapies, resulting in the rapid growth of research output concerned with nature-inspired nanomedicines for cancer immunotherapy. These available nanomedicines are reviewed in the sections below.

### 3.1. Lipid-Based Nanomedicines

#### 3.1.1. Liposomes

Liposomes have been widely explored since 1965 owing to their biocompatibility, biodegradability, simplicity of preparation and surface modification, and diversity of administration routes [99]. The particle size of liposomes usually ranges from 20 nm to more than 1 μm, the proper range of which can promote their infiltration in tumor sites. In addition, their structure enables them to entrap hydrophilic drugs in the aqueous core or hydrophobic drugs within lipid bilayers, implying the possibility of loading multiple drugs for combination therapy. Based on these advantages, several liposome products (e.g., Doxil^®^ from Sequus Pharmaceuticals in USA, Myocet^®^ from Elan Pharmaceuticals in Mumbai, and Taxol^®^ from Aphios Pharmaceuticals in USA) have been approved for cancer therapy to enhance targeting capability and reduce off-target toxicity, revealing the great potential of liposomes in clinical transformation [100,101].

Novel applications of liposomes in cancer immunotherapy are blooming with diversified loading strategies and combination preparations emerging (Table 1) [102,103]. For example, chemotherapeutics are usually used in combination with ICI or modulators to induce ICD and further improve the efficacy of ICB immunotherapy, for which liposomes can provide a multifunctional drug delivery platform to realize a combination of multiple drugs. For instance, two different liposomes were used for co-delivery of DOX and 5-carboxy-8-hydroxyquinoline (IOX1), in which IOX1 downregulated PD-L1 and inhibited P-glycoproteins (P-gp) of cancer cells, thus greatly upregulating effective concentration of DOX and enhancing DOX-induced ICD with reduced tumor immunosuppressive factors [21]. DOX-encapsulated liposomes were also co-loaded with anti-PD-L1 antibody in the gel for the elimination of postoperative immunosuppression through multi-step activation of the cancer-immunity cycle [104]. In addition to single drug-loaded liposomes, two-in-one liposomes have become more predominant for combination immunotherapy through the co-delivery of two chemotherapeutics or chemotherapeutics with adjuvants [105]. For example, Wang et al. constructed neurotransmitter analogs-modified liposomes for co-delivery of camptothecin (CPT) and Cur to glioma and found that Cur could effectively downregulate the CPT-induced elevated expression of PD-L1 and thus prevent the inactivation of T cells and achieve enhanced chemo-immunotherapy [106]. Such a dual-drug combination compensates for the negative effects of a single drug from the mechanism and thus achieves initiation of the self-sustaining cancer-immunity cycle. How to amplify this advantage by adjusting the proportion and release sequence of multiple drugs within an intelligent liposome platform might be an important direction of future research.

The off-target delivery of chemotherapeutics is prone to toxicity issues such as hepatotoxicity and cardiotoxicity, which is appealing for more efficient and accurate tumor-targeted delivery. Surface modification with specific ligands has been an invaluable tool, in which the targeted groups are conjugated with lipids and then embedded in the phospholipid bilayer. Distearoyl phosphoethanolamine-polyethylene-glycol (DSPE-PEG)-maleimide is a common commercial functional phospholipid that can be covalently coupled with a tumor-selective antibody (such as an anti-CD44 monoclonal antibody) or cell-penetrating peptides (such as internalizing-RGD (iRGD)) by Michael addition, thus endowing liposomes with active targeting capability and leading to enhanced accumulation of liposomes at tumor sites [107,113]. Cholesterol-mediated ligand modification is another predictable strategy for liposome functionalization [114]. A DOX-loaded liposome (DOX-SAL) modified with a sialic acid (SA)-cholesterol conjugate (SA-CH) has been developed to facilitate the capture and cellular uptake by tumor-associated macrophages (TAMs). The DOX-SAL NPs dramatically relieved TAM-mediated immunosuppression and promoted CD8+ T-cell-mediated immune response. Moreover, the existence of cancer stem cells (CSCs) involves the immunosuppressive tumor microenvironment, resulting in the propagation and metastasis of tumors [115]. To eradicate CSCs and further improve treatment efficacy in an immunologically “cold tumor”, Li et al. also utilized metformin (MET) (a natural compound derived from the legume Galega officinalis) as a CSC-targeting agent and co-loaded it with DOX into SA-modified liposomes, of which the combination therapy with aPD-1 mAb showed increased benefit on both primary tumor inhibition and metastasis suppression [116,117]. Furthermore, TME-responsive liposomes containing natural compounds have also been widely explored to achieve tumor-specific and locally activated immune responses at tumor sites in a safe manner. Within these liposomes, drugs often exist in the form of prodrugs that can respond to the signals in the TME (e.g., low pH, reductive environment, high concentration of ROS, and overexpressed enzymes such as matrix metalloproteinases-9 (MMP-9)) and release an active compound to amplify the immune response [110,112]. In addition, empty liposomes containing DSPE-PEG-maleimide (L-Mals) have been also applied as capturing agents to bind to the DOX-induced tumor-derived protein agents by forming stable thioether bonds and promote the subsequent uptake by APCs such as DCs, resulting in infiltration of more CD8+ T cells and fewer Tregs, which also provides a new insight into liposome utilization.

Overall, liposome-based nanomedicines have achieved rapid progress for natural compounds in cancer immunotherapy. Though the microfluidics technique has been developed as an alternative to traditional preparation techniques (e.g., film dispersion method and extrusion) to better control the size distribution and drug encapsulation profile of liposomes, their clinical transformation still faces some challenges, such as low stability, a short cycle half-life, the ease of lipid oxidation and hydrolysis, drug leakage, high production costs, and sterilization problems, which are yet to be addressed.

#### 3.1.2. High-Density Lipoproteins (HDLs)

HDLs are endogenous particles composed of phospholipids and apolipoproteins, which function as vehicles transporting biomolecules (e.g., proteins, genes, and hormones) in vivo [118]. Inspired by their unique structure (a hydrophobic core and a hydrophilic shell) and biological functions, HDLs have been explored as carriers for drug delivery through various preparation techniques (e.g., direct isolation from plasma, disassembly-reassembly strategy, and microfluidics) and have shown ultrasmall size, good biocompatibility, stability, and prolonged circulation [119]. These characteristics make HDLs a multifunctional nano platform for the intratumoral delivery of natural compounds.

While the hydrophobic core of HDLs provides enough reservoirs for encapsulation of hydrophobic drugs, the amphiphilic shell provides anchoring sites for amphiphilic drugs or can be covalently modified with drugs on the phospholipids or apolipoproteins, thus enabling HDLs to adapt to different drugs with physicochemical properties. Chemo-immunotherapy represents a powerful weapon in the fight against cancer by co-delivery of chemotherapeutics and immunoadjuvants, which can induce potent ICD at tumor sites in a safe manner. Han et al. prepared biomimetic HDLs by using the thin-film dispersion method and mixed them with AS1411 aptamer-CpG fused sequence-conjugated DSPE (Apt-CpG-DSPE) to achieve surface anchoring, and DOX was intercalated into the reservoir of nucleotides for the construction of immune HDL nano drugs [120]. Though the biomimetic HDL NPs reconstituted in vitro function as an alternative with similar characteristics to the native one, they are still considered impossible to fully replicate the structural characteristics and biological functions of native HDLs. Hence, the same assembly strategy also applies to native HDLs that are extracted from Cohn’s F-IV [121]. The HDLs co-loaded with DOX and immunoadjuvants could specifically target tumor cells through recognition of surface scavenger receptor class B type I (SR-BI), followed by internalization of CpG-DOX into tumor cells, DOX-induced TAAs release, and CpG-triggered antigen presentation by DCs, as well as potent anti-tumor T cell response. The limited source and complex extraction process are the two major issues that need to be considered when using native HDLs. It is also noteworthy that the expression of HDL-specific receptors varies among cancer cell types, which might be one of the potential factors of HDL-mediated intratumoral delivery. In addition, synthetic mimetic peptides that retain structural similarities of prototype apolipoproteins possess the advantages of easy synthesis, low cost, and functional modification, compared to apolipoproteins [122]. Kuai et al. prepared HDL-like nanodiscs composed of apolipoprotein A-I mimetic peptides and phospholipids and incubated the nanodiscs with lipid-DOX to construct sHDL-DOX with an ultrasmall size of ~10 nm (Figure 4A,B), in which the DOX molecule was conjugated to the phospholipid through a pH-sensitive linker that specifically broke and released DOX under acidic conditions (pH 5) (Figure 4C) [123]. The ultrasmall particle size and extended pharmacokinetics facilitated the intratumoral penetration of sHDL-DOX, which then triggered the ICD of cancer cells for the initiation of the immunologic cascade (Figure 4D), leading to an upregulated “eat me” signal and enhanced potent responses of CD8+ T cells (IFN-γ + CD8+ and AH1-specific CD8+ T cells) (Figure 4E,F). Combination therapy with anti-PD-1 plus DOX-carrying nanodiscs resulted in complete tumor regression in 80 to 88% of murine tumor models, revealing the effectiveness of sHDL-DOX. Recent studies on tumor lipid metabolism indicate that cancer cells require large amounts of lipids and cholesterols for proliferation or other activities. This might be beneficial for efficient HDL-mediated intratumoral delivery. Besides, due to HDLs’ native regulation of lipid metabolism, HDLs might serve as a potential carrier for remodeling the lymphocyte immune dysfunction caused by abnormal lipid metabolism in the tumor microenvironment.

Taken together, the potential advantage of HDLs or HDL-like nanoparticles as vehicles remain underexplored but are promising for cancer immunotherapy. HDLs have been extensively studied in other diseases and applied for precise co-delivery of small-molecule drugs, genes, proteins, and photosensitizers, which guides future HDL-based research concerned with the combination of natural compound-mediated immunotherapy and other therapies. Despite these perspectives, the clinical translation of HDLs still requires more effort and the development of technologies for scale production and quality control.

#### 3.1.3. Nanoemulsions

Nanoemulsions are colloidal systems with sizes ranging from 10 to 1000 nm, of which the main components include the aqueous phase, oil phase, and emulsifying agents. According to the relative distribution of water (W) and oil (O) phases, nanoemulsions can be categorized as biphasic nanoemulsions (water-in-oil W/O or oil-in-water O/W) and multiple nanoemulsions (O/W/O or W/O/W) [124]. Owing to their optical clarity, large surface area, easy preparation, and biodegradability, nanoemulsions such as O/W nanoemulsions have been considered common vehicles for encapsulating hydrophobic natural compounds in the internal phase to improve their solubility and stability [125]. W/O/W nanoemulsions can provide double reservoirs for encapsulating both hydrophobic and hydrophilic compounds for combination therapy but are limited by their instability during storage [126]. This matter can be improved by selecting proper emulsifiers (e.g., proteins, polysaccharides, and surfactants) and preparation technologies (e.g., ultrasound, high pressure, microfluidization, and rotor-stator-based homogenization) [127].

The combination therapy of ICI and chemotherapeutics is a promising strategy for cancer immunotherapy, in which ICI and chemotherapeutics are required to function at different sites of tumors. To realize specified spatial co-delivery of ICD-inducer (DOX) and ICP inhibitor (HY19991 (HY)), a pH-responsive Pickering nanoemulsion was developed by using multi-sensitive nanogels as an oil/water interfacial stabilizer [128]. Due to the pH-responsive hydrophilicity-hydrophobicity switch of nanogels, the nanoemulsion (D/HY@PNE) disassembled to release HY and DOX-loaded nanogels, leading to deep tumor penetration and effective ICD response. However, ICD inducers tend to simultaneously up-regulate the expression of PD-L1 in tumor cells and thus inactivate T cells through the PD-1/PD-L1 pathway, which requires them to be used in combination with PD-1/PD-L1 blockade. Celastrol (CEL) extracted from tripterygium can not only induce potent ICD to activate DCs and T cells but also interrupt the PD-1/PD-L1 pathway, triggering a strong anti-tumor immune response. To improve the solubility and bioavailability of CEL, Qiu et al. prepared CEL-loaded O/W nanoemulsions using the ultrasonic emulsification method [129]. CEL was loaded in the sesame oil core surrounded by soybean lecithin and Pluronic F68 with an encapsulation efficiency of (90 ± 2)%. Intratumoral injection of CEL nanoemulsions activated the immune system and inhibited both the treated and the distant untreated tumor in vivo.

Overall, nanoemulsions contribute to the improvement of solubility, stability, release profile, and bioactivity of natural compounds. It is noteworthy that strict quality control and improved system stability should be emphasized to promote their large-scale production. Novel nanoemulsion-based nanomedicines are yet to be explored, as well as technical innovation.

### 3.2. Micelle-Based Nanomedicines

Micelles are generally formed by self-assembly of amphiphilic copolymers when over a certain concentration (referred to as critical micelle concentration). While the hydrophobic moiety of polymers gathers to form a core, the hydrophilic moiety is oriented toward the external polar environment to form a shell that functions as a barrier protecting the internal groups and preventing micellar aggregation and thus maintains good stability in water.

As a self-assembled system, micelles provide a promising platform for drug delivery, especially for applications of natural compounds in cancer immunotherapy. Micelles with a size range of 10–150 nm possess durable plasma circulation, as well as good tissue penetrability and endocytosis by cancer cells [130]. Furthermore, the amphiphilic structure provides diverse loading sites for natural compounds with different structures, in which the hydrophobic payloads (e.g., PTX, DOX, CEL, and Cur) can be easily encapsulated into the core in prodrug or active forms, while the hydrophilic payloads need to be conjugated with hydrophobic moieties to endow them with the feasibility of encapsulation within the core during self-assembly, or to be incorporated onto the surface of micelles through non-covalent interaction [22,131]. This characteristic makes micelles interesting dual-drug-loaded vehicles to synergistically enhance anti-tumor immune response [132]. A micelle based on the chimeric cross-linked polymersome (CCPS) has been developed for the co-delivery of low-dose DOX to induce ICD and a photosensitizer to facilitate PDT, leading to enhanced DC recruitment and an amplified immune response cascade [133]. The CCPS containing primary and tertiary amines also functioned as an adjuvant that formed an in situ DC vaccine when combined with TAAs. In recent years, strategies that combined two different micelles each loading two drugs (DOX and shMFN1, Cur and CpG-ODN) have also been proposed to selectively target cancer cells to trigger ICD and target DCs or macrophages to initiate an immune response [134,135]. Such a dual-micelle strategy focuses on regulating different target cell populations within the tumor tissues, in which the drug combination, administration dosage, and the order of administration should be further considered to achieve optimal efficacy.

The multifunctionalization of polymers facilitates the development of multifunctional micelle. The introduction of targeted fragments on the surface of micelles promotes the uptake of target cells or adhesion on leukocytes to achieve leukocyte-hitchhiking delivery for a higher accumulation in tumor tissues [136,137]. To achieve programmed drug release that acts on different targets, tumor-microenvironment-responsive micelles have been developed for natural compound delivery and TIME reprogramming [138,139,140]. For example, a pH/redox cascade-responsive micelle has been constructed that can shrink sizes and convert to a positive charge in responding to weak acidity in tumor tissues, resulting in enhanced tumor penetration and endocytosis followed by the release of Cur and NLG919 (an immune checkpoint inhibitor for IDO) in redox-rich cytoplasm [141]. Based on the different acidity characteristics of the tumor extracellular microenvironment (approximately pH 6.5) and lysosomal environment (approximately pH 5.5), dual-pH-responsive micelles were reported to achieve sequential release of two drugs acting on extracellular (anti-PD-1 mAbs to block PD-1 in T cells) and intracellular (Cur to promote the tumor infiltration of T cells by inhibiting NF-κB pathway), respectively [142]. Wang et al. also constructed dual-pH-responsive micelles (SRNs) for sequential delivery of two different prodrugs (ICD inducer and DC stimulator) (Figure 5A) [143]. SRNs accumulated in both tumor tissues and lymph nodes tissues (Figure 5C,D), in which NPs responded to signals of TME (pH < 6.9 and high-level glutathione (GSH)) to release DOX early in tumor tissues to trigger ICD, while the remaining NPs were phagocytized by DCs in sentinel lymph nodes tissues and released imidazoquinolines to promote DC maturation (Figure 5B,E), providing a potential micelle-based candidate to advance synergistic cancer chemo-immunotherapy. This sequential delivery strategy conforms to the sequence of initiation and boost of the cancer-immunity cycle in theory. In this regard, it might be meaningful for better amplification of anti-tumor immune response.

Taken together, micelles possess far-reaching research value and broad prospects in the tumor-targeted delivery of natural compounds for cancer immunotherapy. However, the preparation conditions of micelles are relatively strict, especially for their concentration. The interaction of micelles with the in vivo environment is complex and the mechanisms involved need to be further revealed.

### 3.3. Polysaccharide-Based Nanomedicines

As a common biomacromolecule in nature with abundant sources, good biocompatibility, excellent stability, and low immunogenicity, polysaccharides (e.g., alginate, chitosan, and cyclodextrin) have also been exploited as polymer materials for natural compound delivery [144,145].

Polysaccharides contain a variety of active groups, such as amines (e.g., chitosan), amides (e.g., chitosan and hyaluronic acid), and the hydroxyl group (e.g., cyclodextrin and hyaluronic acid), which can be covalently linked for further modification [144]. For example, grafting with hydrophobic materials allows the tuning of physicochemical properties such as solubility or amphiphilicity, improving their abilities to encapsulate drugs and assembly behavior. In addition to being encapsulated within systems, drugs can also be covalently linked to the saccharide chain. For example, cyclodextrins (CDs) are natural products with a hydrophobic cavity and hydrophilic shell, providing enough reservoirs to encapsulate hydrophobic cargoes through guest-host recognition. Yang et al. prepared a supramolecular monomer (CD-S-CPT) by conjugating the natural compounds (CPT) to β-CD via a ROS-cleavable linker [146]. The host-guest complexation and supramolecular polymerization with iron-carboxylate coordination contribute to a 286-fold enhancement in solubility of CPT (Figure 6A). The supramolecular polymers assembled to form supramolecular nanoparticles (SNPs) with uniform sizes (approximately 122 nm) driven by iron-carboxylate coordination. The SNPs with uniform distribution of Fe and S elements could accumulate in tumor tissues (Figure 6B), in which the iron ions could catalyze the production of toxic hydroxyl radicals and further cleave the thioketal linker to released CPT, thus triggering ICD and providing a cascade-amplified strategy for cancer immunotherapy when in synergy with ICI (Figure 6C,D). Moreover, targeting ligands or stimuli-sensitive elements is also encouraged via a covalent modification to achieve tumor targeting and tumor-specific drug release [147,148]. Some polysaccharides such as hyaluronic acid possess an intrinsic tumor-targeting ability. For example, Gao et al. synthesized prodrugs by conjugating DOX to HA via an MMP-sensitive linker (CPLGLAGG) [149]. The drug-linked HA was fabricated into NPs that demonstrated tumor targeting and enzyme-activated DOX release, leading to initiation of an immune response and improved anti-tumor effect when combined with ICI.

The potential advantages of polysaccharide-based nanomedicines remain to be further explored. It is worth noting that their clinical translation seems to be challenging because of the uncontrollability of material parameters (e.g., monosaccharide composition, molecular weight, and charge density) that vary with the source material and processing, thus causing variations in quality. In this respect, the clinical translation of polysaccharide-based nanomedicines requires more advanced extraction and preparation technologies for salable production and quality control.

### 3.4. Peptide-Based Nanomedicines

The peptide self-assembly system is perhaps one of the most common strategies for drug delivery with the advantages of easy synthesis, flexible functionalization, and customizable physicochemical properties, in which the compounds can be non-covalently encapsulated in peptide systems or chemically conjugated to peptides. Ascribed to the 20 different amino acids to be selected, the diversified functions of peptide-based nanosystems are also easy to achieve by fusing peptide motifs with different biological activities (e.g., tumor targeting, molecular recognition, and signal transduction) [150]. Moreover, some anticancer peptides such as melittin peptides can be introduced by a peptide fusion strategy and thus play a synergistic anti-tumor effect with natural compounds with reduced immunosuppression and increased innate immunity [151].

While the peptide materials self-assemble into a variety of forms (e.g., particles, fibers, and gels) by noncovalent intermolecular forces (e.g., hydrophobic interaction, intermolecular π-π stacking, hydrogen bonding, and electrostatic force), the changes in the involved forces also lead to structural transformations, thus promoting them as stimuli-sensitive devices for tumor-specific drug release [152]. The introduction of linkers that respond to the signals of the TME (e.g., pH, ROS, GSH, and enzymes) or external stimulus (e.g., light and heat) into the peptide nanosystems has gained great attention for realizing an “on-off” drug release pattern for cancer immunotherapy [153,154,155]. For example, the photothermal performance of the photosensitizer (Ag_2_S quantum dots) loaded in a peptide hydrogel (PC_10_A–arginine–glycine–aspartic acid) could promote the phase transition of hydrogel from solid to liquid and thus trigger the sustained release of DOX to initiate local vaccination in tumor sites [156]. To achieve targeted delivery of drugs to tumor sites, relieve immunosuppression, and elicit an immune response, various enzyme-sensitive systems have been also reported. Sun et al. designed amphiphilic biofunctional peptides using a fibroblast activation protein-α (FAP-α) responsive peptide to conjugate both PD-L1 and PD-1, and encapsulated two ICD inducers including DOX and R848 to construct PCP@R848/DOX [157]. The prodrug nanoparticles were disassembled by FAP-α and specifically released drugs, efficiently propagating the activation of cytotoxic T cells and the anti-tumor immune response. These results reflect that co-assembly of different functional peptide fragments contributes to the combination of multiple functionalities for improved immunotherapeutic or diagnostic performance. In addition, the combination of peptide nanomaterials with synthetic amino acids (such as D-type amino acids) or adjuvant-like polymers might also be a potential strategy, in which the compatibility between different materials and the treatment needs of certain tumor types need to be considered during design.

Overall, peptide-based nanomedicines have made much progress. Though they have advantages of simple synthesis and assembly, their formulation stability is difficult to maintain and strict preservation conditions are usually required. Based on the motifs that determine the property and functionality of peptides, novel motif development or suitable motif compatibility are yet to be explored.

### 3.5. Protein-Based Nanomedicines

Owing to their excellent biocompatibility, biodegradability, and various functional groups (e.g., amino, thiol, and carboxylic groups) for modification, several animal-derived proteins (e.g., serum albumin, ferritin, gelatin, and collagen) have been explored for drug delivery with the development of preparation technologies (e.g., emulsion and solvent extraction, electrospraying, and salt precipitation) [158,159]. Albumin-based nanomedicines, as representative examples, have been approved by the FDA and taken the lead in clinical practice to treat breast, lung, and pancreatic cancer, highlighting the great prospects of therapeutics in this category [160].

Albumin as the most abundant protein in the blood has a long half-life (~19 days), transendothelial capacity via GP60-receptor-mediated transcytosis, and several applicable administration routes (e.g., intravenous injection and inhalation), which make it an excellent carrier for drug delivery. Co-delivery of natural compounds and other immunomodulators using albumins as vehicles has been reported for cancer immunotherapy [161]. For example, PTX-loaded albumin nanoparticles that were conjugated with anti-PD-L1 mAbs through a pH-sensitive linker showed enhanced tumor accumulation, effector T cell infiltration, and regulatory T cell suppression with low organ toxicity [21]. In addition, Hu et al. reported hyaluronan-coated albumin NPs for the co-delivery of a hydrophobic natural compound (celastrol) and hydrophilic IDO inhibitor (1-methyltryptophan), achieving effective tumor inhibition in pancreatic cancer mouse models by relieving the immunosuppressive TME [162]. Transferrin is the main iron-containing protein in plasma. Owing to their innate functions of transporting cargoes and targeting tumor tissues mediated by transferrin receptor 1, which is overexpressed in cancer cells, transferrin has also emerged as an excellent vehicle for therapeutic encapsulation and delivery. Zhang et al. prepared PTX-loaded transferrin (PTX@TF) and then co-encapsulated it with marimastat in thermosensitive liposomes (TMNP) [163]. Triggered by hyperthermia treatment at tumor sites, the PTX@TF was rapidly released and entered cancer cells, creating synergies with marimastat, thus leading to improved tumor infiltration of CD8+ (4.2-fold) and CD4+ T cells (1.7-fold). In addition, it has been reported that caged proteins are well-suited for cancer immunotherapy because of their virus-like geometries and surface properties, which promote the passive delivery of tumor-specific antigens in the lymphatic systems followed by antigen presentation and immune cell activation. Viral or nonviral protein cages have been widely explored and engineered to elicit immune responses for cancer therapy. Though there are still few studies on their delivery of natural compounds, caged proteins possess great potential in this respect based on their passive targeting of lymph nodes and/or viral protein cage-related immune activation capacity [164].

Taken together, protein-based nanomedicines hold great promise in delivering natural compounds for cancer immunotherapy. However, the high cost of recombinant protein synthesis and production hinders their further large-scale application. Moreover, the modification is generally restrained by the structure of proteins, especially when it needs to maintain their biological activities. In addition, the shelf-life of protein-based formulations is usually short owing to their easy degradation.

### 3.6. Cell-Derived Nanomedicines

#### 3.6.1. Living Cell-Mediated Nanoparticles

Living cells with inherent chemotaxis are natural “living therapeutics” that can navigate among physiological barriers and participate in tissue repair in vivo and have emerged as promising therapeutics with some encouraging results obtained from clinical trials. Inspired by their particular physiological nature in terms of good biocompatibility, tumor tropism, long circulation, and immune response, living cells (e.g., neutrophils, DCs, macrophages, T cells, and platelets) have also been applied as promising drug carriers with non-duplicated and incomparable advantages over traditional nanomedicines (Table 2), such as high intratumoral accumulation and potent anti-tumor immune response [165]. There are two main drug-loading strategies: the drugs or nanoparticles can be modified on the cell surface through cell-surface engineering or loaded into cells through the endocytosis pathway. The choice of drug-loading pattern depends on the physiological–biochemical characteristic of cells and the properties of drugs or nanoparticles, as well as the demand for drug delivery and release.

The efficacy of the infused immune cells (e.g., DCs and CD8+ T cells) is nonetheless undermined by immunosuppression at tumor sites. To solve this problem, Li et al. proposed a strategy using DCs containing ICD inducer-loaded NPs (Nano-DOX-mDC) [167]. Nano-DOX-mDC effectively infiltrated into the glioma after intravenous injection. The released DOX stimulated local immunogenicity to promote DC-mediated antigen presentation, eliciting an anti-tumor immune response. The infused autologous immune cells are also easily domesticated into tumor-promoted phenotypes and need extra processing. For example, macrophages play a vital role in the occurrence, development, and metastasis of tumors, in which M1 macrophages destroy cancer cells, while M2 macrophages such as tumor-associated macrophages (TAMs) promote tumor growth and invasion. Control of the infused subtypes of macrophages and repolarization of tumor-associated macrophages from the M2- to M1-like phenotype are promising approaches to address this issue [173]. Wang et al. developed celastrol NP-loaded M1-like macrophages, in which the M1-like macrophages served as both carriers that encapsulated celastrol and living therapeutics that released a serious of anti-tumor cytokines (e.g., IL-1β, IL-6, and TNF-α) [166]. While the celastrol NPs maintained the polarized status of M1-like macrophages and intervene in M2-like polarization, the exocytosed celastrol NPs with tumor cytotoxicity synergistically enhanced the anti-tumor effect. Other immune cells such as neutrophils are also applied to deliver liposomes into the brain for glioblastoma treatment. Li et al. prepared a nanosensitizer (ZGO@TiO_2_@ALP) containing a ZnGa_2_O_4_:Cr^3+^ core for luminescence imaging and sono-sensitive TiO_2_ shell for ROS-responsive controlled release of interior anti-PD-1 antibody and PTX loaded in liposomes [168]. Neutrophils mediated high BBB penetration and tumor-specific accumulation of nanosensitizer and achieved augmented and sustained therapy under the stimulus of ultrasound irradiation, providing a novel platform for precision glioblastoma immunotherapy. It is worth noting that the level and state of immune cells in each patient are different. Hence, relevant in vivo detections about NKs, CTLs, and other immune cells should be carried out before the treatment and thus comprehensively understand the original level of immunity in the body. On this basis, a suitable treatment plan containing certain type of immune cells could be proposed. In addition to immune cells, other living carriers are also utilized for the precise delivery of immunomodulators. Based on the tumor tropism and responsive microvesicle-secreting capabilities of platelets, Lu et al. proposed engineered platelets with inner loading of ICD inducer DOX and external anchoring of aPD-L1-cross-linked nanogels through covalent modification, resulting in T-cell-mediated tumor cell death and reversion of immunosuppressive microenvironment in tumor tissues [170]. In addition, other cells such as mesenchymal stem cells, red blood cells (RBCs), and even some bacteria with tumor tropism possess relatively abundant sources and thus also have high potential to deliver natural compounds for cancer immunotherapy in the future [154,174,175].

Despite these remarkable achievements of living cell-mediated nanoparticles for cancer immunotherapy, there are still several challenges to be addressed before clinical translation. During the preparation process, the loading method should not affect the cellular activity and biological functions and thus further studies are needed to optimize the production process. In addition, comprehensive methods are needed to evaluate the in vivo behaviors of living cells, such as distribution, kinetics, and metabolism, and their survival in the lesions, thus guiding their use safely and efficiently.

#### 3.6.2. Cell Membrane Camouflaged Nanoparticles (CM-NPs)

Cell membranes, as the important interfacial structure between cells and the external environment, provide great support for various cell biological activities. This characteristic prompted the emergence of cell-membrane-coating technology and researchers directly leverage cell membranes instead of synthetic techniques to cloak nanoparticles, thus bestowing them with excellent cell-mimicking functions and augmented potency and safety. There are several extant cell membrane types, including red blood membrane, platelet membrane, leukocyte membrane, cancer cell membrane, and stem cell membrane [176]. In addition, cell-membrane-coating technology has broad applicability for encapsulating various inner core nanocarriers, such as polylactic-co-glycolic acid (PLGA), MSNs, iron oxide, metal-organic frameworks, and liposomes [177]. Components (cell membranes and inner core materials) should be selected according to the needs of specific drug delivery and the purpose of treatment.

CM-NPs fuse the advantages of both the host cells and the nanomaterials and show great potential in tumor-targeted delivery of natural compounds for cancer immunotherapy. For example, a cancer membrane not only acts as a functional cloak that endows NPs with high stability, long circulation, tumor-oriented homing, and self-recognition but also functions as a cancer vaccine with multivalent tumor antigens. Wan et al. extracted a cancer cell membrane from a drug-resistant lung carcinoma cell line and developed a cell membrane vehicle for the co-delivery of DOX and sorafenib [178]. The hybrid NPs showed strong accumulation in the tumor tissues at 4 h post-injection with reduced distribution in the reticuloendothelial system. Furthermore, the DOX induced significant ICD with sorafenib and remodeled the TME with downregulation of Treg, activation of effector T cells, and relief of PD-1 expression. Inspired by the recruitment and infiltration characteristics of immune cells (e.g., macrophages and T cells) at tumor sites, they represent another source of membrane coating materials to promote circulation and tumor targeting of NPs [179,180]. The components (e.g., lipid bilayer and membrane proteins) in the membrane also provide various sites for further modification, which facilitate the combination of cell-membrane-coating technology and membrane modification technology to develop a multi-function platform of CM-NPs. For instance, phenylboronic acid (PBA) covalently bonded to the membrane protein of the T cell membrane [181]. Then the PBA-modified T cell membrane covered Cur-loaded redox-sensitive HA NPs through the boronic ester dynamic covalent bond between PBA and glycosyl. The coating membrane dissociated in the acidic TME to release Cur-loaded NPs for tumor chemotherapy and the cell membrane debris for sheltering the PD-L1 of tumor cells. As for some functional moieties (such as CXCL4 for enhanced chemotaxis) that are difficult to directly modify on the extracted cell membranes, cells can be pretreated using genetic engineering or glucose metabolism engineering technology, followed by membrane extraction. More functional membrane modification strategies and drug combination strategies need to be further explored.

Although cell-membrane-camouflaged nanoparticles possess clear advantages, much work needs to be completed before their application in clinical practice. Current low extraction efficiency still requires relatively plentiful materials from donors for blood products (e.g., RBCs, platelets, and leukocytes) and large-scale culture for other cell types. Strict attention should also be paid to maintaining the activity of biological components in the membrane during preparation, especially when additional modifications are introduced for the construction of multi-functional intelligent nanoplatforms.

### 3.7. Carrier-Free Nanomedicines

Carrier-free nanomedicines that are completely self-assembled or co-assembled by active drug molecules have aroused scientists’ attention and been developed as an alternative modality to overcome the drawbacks of traditional nanomedicines such as limited drug-loading efficiency (less than 10%) and carrier-related toxicity [182]. The interaction between drug molecules within carrier-free nanoparticles involves non-covalent bonds (e.g., hydrophobic interaction, intermolecular π-π stacking, hydrogen bonding, and electrostatic force), resulting in extremely high or even almost 100% drug-load efficiency. Their simple and reproducible preparation techniques (e.g., one-step nanoprecipitation method, anodized-aluminum oxide (AAO) template-assisted method, and ice-template-assisted method) are of benefit to their quality control during large-scale production and clinical transformation [183,184].

In recent years, some natural small molecules such as pentacyclic triterpenoids have been revealed to have self-assembly properties, resulting in rapid innovation in the construction of carrier-free nanoparticles based on natural compounds. For example, ursolic acid (UA) is a plant-derived pentacyclic triterpenoid carboxylic with immunostimulatory activity and low toxicity. To address its issues, including poor water solubility, low specificity, and poor bioavailability, Fan et al. prepared a carrier-free pure nanoparticle (UA NPs) by pure single UA self-assembly via intermolecular forces based on electrostatic and hydrophobic interaction [185]. The carrier-free UA NPs exhibited the improved immunostimulatory activity of TNF-α, IL-6, and IFN-β in vitro and inhibited tumor growth with significant activation of CD4+ T cells in vivo, revealing their great potential for cancer immunotherapy. Another self-assembly of the single natural compounds with anti-tumor activity has also been investigated, including Cur, CPT, and PTX. Nanomedicines constructed using this self-assembly approach based on a single drug are prone to aggregation or dissolution because of the imbalanced intermolecular interactions. Co-assembly of dual drugs that reaches an equilibrium of intermolecular interactions, by contrast, have shown distinct advantages in assembly stability and co-delivery of two different drugs, thus highlighting their potential in anti-tumor combined therapies through targeting different pathways synergistically. For example, compounds with similar structures (such as several pentacyclic triterpenoids) have the potential for construction of the co-assembled NPs that maintain the biological activity of individual units, thereby exerting a synergistic therapeutic effect for cancer treatment [186]. Pentacyclic triterpenoids are also used as bioactive carrier materials for loading other hydrophobic drugs. For instance, UA and PTX could be co-assembled into nanocomposites (UA-PTX NPs) through hydrogen bonding and hydrophobic interaction [187]. The in vivo experiments revealed that UA-PTX NPs maintained the biological activity of UA, therefore exerting a synergistic effect with PTX and alleviating PTX-induced organ damage. Other natural compounds and their derivatives with self-assembly capability also showed the potential to act as both the carrier and the bioactive synergist with other therapies. For example, Zhang et al. synthesized a Cur-based ROS-response prodrug, which can load with the photosensitizers (PS) through a hydrophobic effect and π−π stacking [188]. In addition to the synergistic therapeutic effect, the red fluorescence of PS and the green fluorescence of Cur with “off-on” features enable the hybrid NPs with real-time self-monitoring of drug distribution and release in vivo, providing a new field of vision to design intelligent carrier-free nanomedicines using multifunctional natural compounds.

Some first-line chemotherapeutics (e.g., PTX and DOX) derived from plants can combat the immunosuppressive tumor microenvironment (ITM) and their related carrier-free nanomedicines using co-assembling molecule pairs (e.g., DOX-CPT and PTX-ICG) that are applied for a combination of immunotherapy with chemotherapy or phototherapy [189,190]. For example, lollipop-like NPs were assembled via π–π stacking and hydrogen bonds for the co-delivery of DOX and gossypol, in which a small amount of polydopamine (PDA) acted as an adhesive that filled the intermolecular gaps and overcame the unstable interaction between DOX (hydrophobic) and gossypol (hydrophilic), achieving high drug loading (91%) and ultralong circulation (>192 h), as well as excellent synergistic therapeutic efficacy [191]. It has been suggested that PTX at a low dose (5 mg/kg) can efficiently combat an immunosuppressive tumor microenvironment (ITM) by reducing intratumoral infiltration of Tregs and suppressing their immune inhibitory function. Recently, Feng et al. reported an ICG-templated self-assembly strategy for preparing two-in-one NPs (termed as ISPN) and combination immunotherapy to treat triple-negative breast cancer [190]. The combination of ISPN and ICI achieved a synergistic anti-tumor effect through three-pronged modulation pathways, demonstrating the influential roles of carrier-free co-assembled NPs in an anti-tumor immune response from multiple links.

Carrier-free nanomedicines based on natural compounds have shown several advantages in drug delivery and tumor immunotherapy for combination therapies. However, there are still several issues to be addressed in future research. First, it is hard to predict whether a molecule can self-assemble or co-assemble into NPs; second, multiple factors (e.g., assembling performance, system stability, and synergistic effect) need to be considered to select the optimal proportions of different components.

## 4. Conclusions and Discussion

In this review, we summarized and elucidated the potential immunoregulatory mechanisms of natural compounds at every stage of the cancer-immunity cycle. We also illustrated the current limitations of natural compounds in cancer immunotherapy and then proposed nanotechnology-based strategies to explore how to promote their development for enhanced cancer immunotherapy. Recent advances in delivering natural compounds for cancer immunotherapy based on various drug delivery systems with different characteristics were summarized and discussed, of which the advantages and disadvantages were also analyzed, thus providing a basis for the future development and clinical transformation of nature-inspired nanomedicine, especially for cancer immunotherapy.

Despite the potential of natural compounds in immunotherapy against cancer and their recent significant strides (e.g., improved solubility and bioavailability, enhanced tumor targeting and penetration, and excellent efficacy through a multidrug combination) based on nanotechnology, there are still many bottlenecks in their applications for cancer immunotherapy, including unclear therapeutic mechanisms, diverse pharmacological activities that are difficult to control, high effective dose, and the complex extraction and purification process. Therefore, more technological innovation and scientific exploration are needed to illustrate the pivotal mechanisms and break through the barrier between their theoretical research and practical applications. It is worth noting that nature offers a treasure trove of compounds, which have yet to be discovered and elucidated by researchers from technical and scientific perspectives. We anticipate that this review could benefit researchers in various fields including pharmacology, natural product chemistry, materials science, and Chinese pharmacy. With the development of these pharmaceutical sciences and the delivery technology, we fully look forward to the far-reaching development of natural compounds in cancer immunotherapy, even into their clinical practice.

## Figures and Tables

**Figure 1 pharmaceutics-14-01589-f001:**
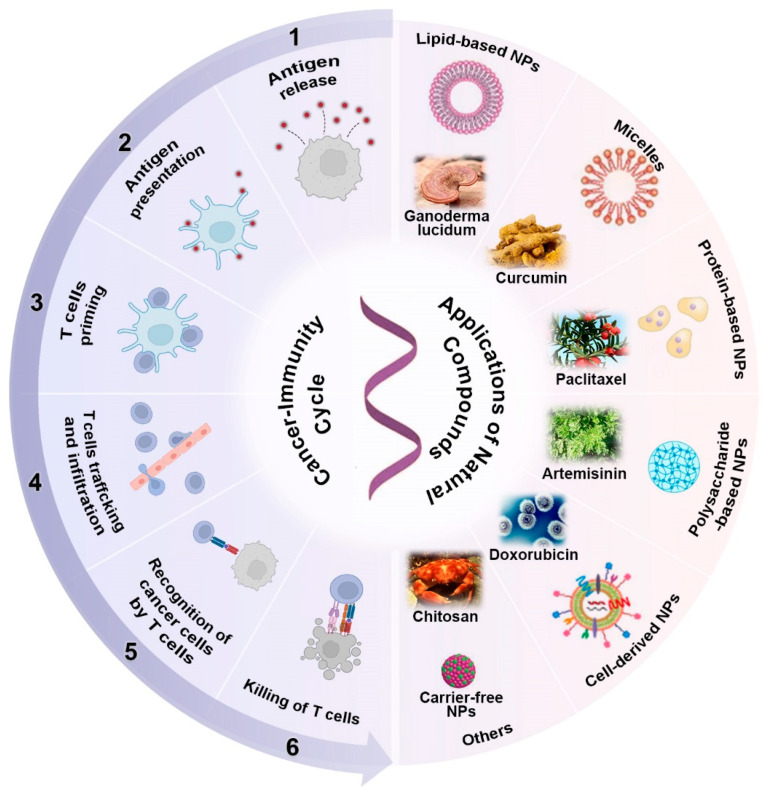
An overview of different steps of the immune response, delivery strategies, and biomedical applications of natural compounds for cancer immunotherapy.

**Figure 2 pharmaceutics-14-01589-f002:**
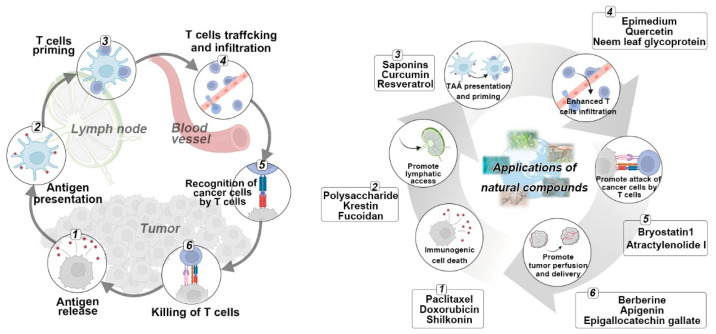
Immune regulatory mechanisms of various natural compounds based on the cancer-immunity cycle.

**Figure 3 pharmaceutics-14-01589-f003:**
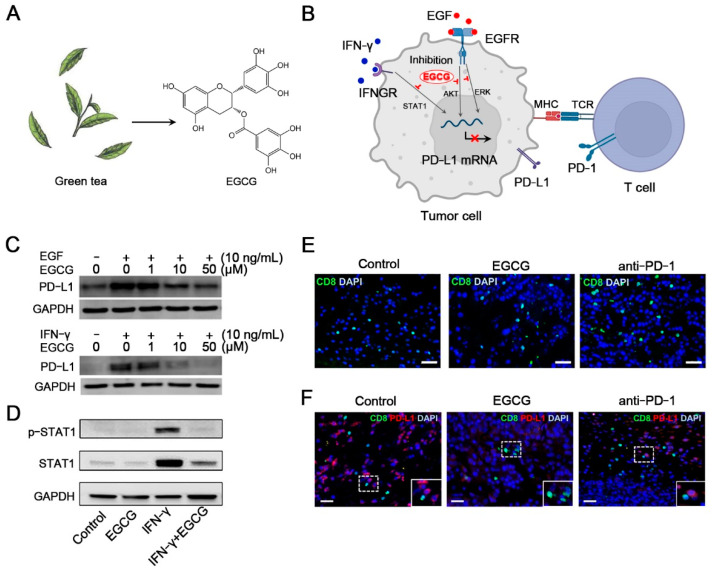
Schematic of EGCG for cancer immunotherapy. (**A**) EGCG is the active compound derived from green tea. (**B**) Schematic of the mechanisms of EGCG for the inhibition of the PD-1/PD-L1 axis. (**C**) Downregulation of epidermal growth factor (EGF) and IFN-γ-induced PD-L1 protein in Lu99 cells after EGCG treatment. (**D**) Downregulation of p-STAT1 and STAT1 in B16F10 cells after EGCG treatment. (**E**) Immunofluorescence images of tumor tissue sections stained with CD8 (green) and DAPI (blue). (**F**) Immunofluorescence images of tumor tissue sections stained with CD8 (green), PD-L1 (red), and DAPI (blue). Reprinted with permission from Ref. [90]. Copyright 2018, copyright MDPI. Reprinted with permission from ref. [91]. Copyright 2021, copyright MDPI.

**Figure 4 pharmaceutics-14-01589-f004:**
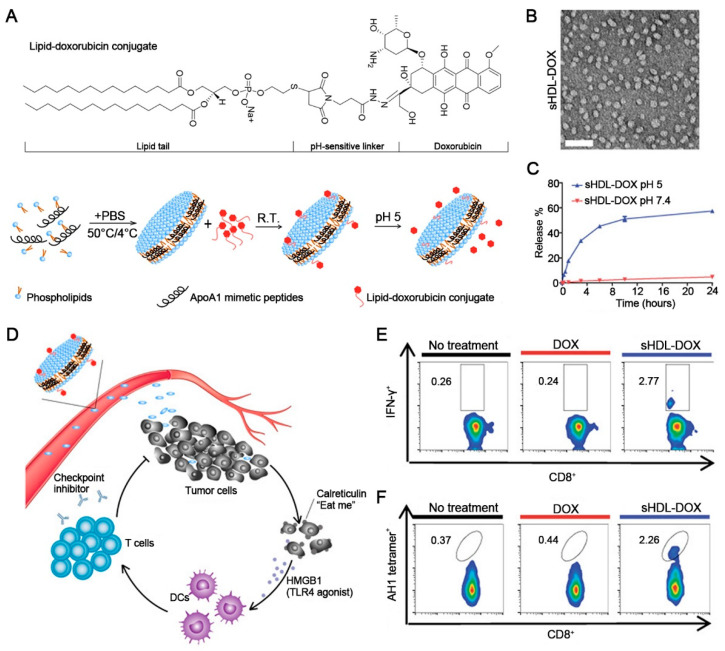
Schematic of sHDL-Dox for chemo-immunotherapy. (**A**) Schematic of the lipid-DOX conjugate and the preparation process. (**B**) The image of sHDL-DOX using a transmission electron microscope. Scale bars, 50 nm. (**C**) The release of DOX at pH 5 and pH 7.5. Data represent mean ± SD (*n* = 3). (**D**) Schematic of sHDL-DOX triggering antigen release for cancer immunotherapy. (**E**) The increased percentage of IFN-γ + CD8+ T cells induced by sHDL-DOX. (**F**) The increased percentage of CT26 tumor antigen peptide AH1-specific CD8+ T cells induced by sHDL-DOX. Reprinted with permission from Ref. [123]. Copyright 2018, copyright the American Association for the Advancement of Science.

**Figure 5 pharmaceutics-14-01589-f005:**
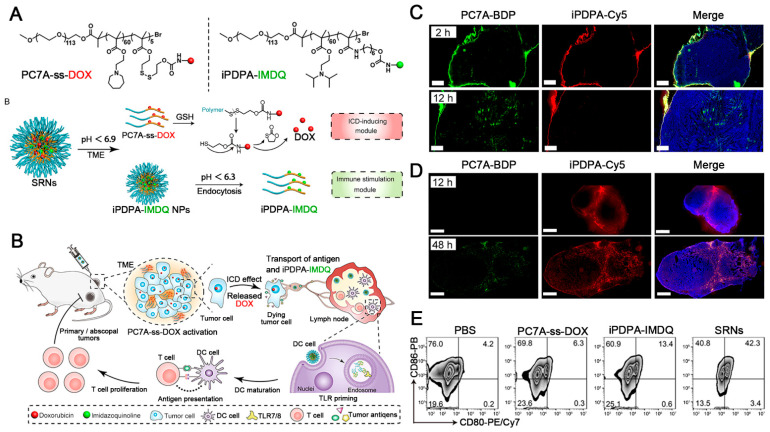
Schematic of SRNs for cancer immunotherapy. (**A**) Schematic of the structure of two prodrugs and the pH-responsive behavior for sequential drug release. (**B**) Schematic of sequential activation mechanisms of immune response after peritumoral injection with SRNs in the tumor-bearing mice. (**C**) Representative images of tumor tissues at 2 and 12 h after peritumoral injection with SRNs. (**D**) Representative images of lymph nodes tissues at 12 and 48 h after peritumoral injection with SRNs. (**E**) Expression levels of CD80 and CD86 on DCs after treatment of SRNs. Reprinted with permission from ref. [143]. Copyright 2021, copyright American Chemical Society.

**Figure 6 pharmaceutics-14-01589-f006:**
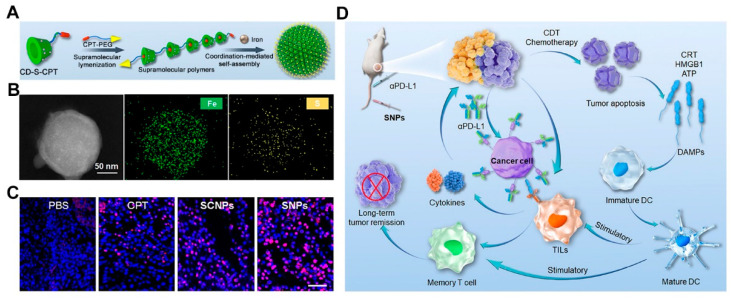
Schematic of SNPs for cancer immunotherapy. (**A**) Schematic of the assembly of the supramolecular monomer (CD-S-CPT) and construction of SNPs. (**B**) The elemental mapping of SNPs. (**C**) Representative images of TUNEL staining of the tumor tissues after treatment with different preparations. (**D**) Schematic of the mechanism of SNPs-based combination therapy with ICI to elicit anti-tumor immunity. Reprinted with permission from Ref. [146]. Copyright 2022, copyright Wiley-VCH.

**Table 1 pharmaceutics-14-01589-t001:** Liposome-based combination preparations for cancer immunotherapy.

Drug Combination	Loading Strategies	Improved Effect	Disease Models	Ref.
DOX and carborane (CB)	DOX-CB conjugates were entrapped in the lipid bilayer	Coupling boron neutron capture therapy with immunotherapy	Glioblastoma models	[107]
DOX and CpG	DOPE-DOX-conjugate and DOPE-MMP-9 responsive peptide-CpG conjugate self-assemble into NPs	Co-delivery of chemotherapeutics with adjuvants	E.G7-OVA tumor models	[105]
PTX and αGC	PTX and glycolipid αGC were co-encapsulated in the lipid bilayer	Co-delivery of chemotherapeutics with adjuvants	B16F10 melanoma xenograft and lung metastasis models	[108,109]
CPT and Cur	CPT and Cur were entrapped in the lipid bilayer	Cur could downregulate the CPT-induced elevated PD-L1 expression and reduce Treg-mediated immunosuppression	Glioblastoma models	[106]
DOX and PD-L1 inhibitor	DOX was encapsulated inside and DSPE-PEG2000-MMP-responsive peptide-PD-L1 inhibitor was inserted into the lipid bilayer	Combination of cancer immunotherapy and chemotherapy to enhance the anti-tumor effect	B16F10 melanoma models	[110]
DOX and silybin (SLN)	DOX-loaded liposomes and SLN-loaded liposomes	SLN-loaded liposomes could change stromal structures and abrogate immunosuppression when in combination with DOX-loaded liposomes	Triple-negative breast cancer	[111]
IOX1 and DOX	IOX1-loaded liposomes and DOX-loaded liposomes	IOX1 could inhibit P-gp of cancer cells to enhance DOX-triggered ICD	Triple-negative 4T1 breast cancer models	[21]
PTX, thioridazine (THZ), and HY19991 (HY)	PTX-loaded micelles, THZ, and HY were entrapped in the aqueous core	Co-delivery of therapeutics against bulk tumor cells, cancer stem cells, and immune checkpoints	Breast cancer models	[112]

**Table 2 pharmaceutics-14-01589-t002:** Representative living cell-mediated nanoparticles.

Cells	Key Characteristics	Therapeutics	Disease Models	Ref.
Macrophage	Tumor targeting; phagocytosis; immunoregulatory effect	CEL NPs	Abdominal metastasis of lung cancer models	[166]
DCs	Tumor targeting; immune response; blood-brain-barrier (BBB) penetration	Dox-polyglycerol-nanodiamond composites	Glioblastoma models	[167]
Neutrophils	Tumor targeting; BBB penetration	PTX and aPD-1 co-loaded NPs	Glioblastoma models	[168]
T cells	Tumor targeting; immune response	Liposomal immune regulators	Subcutaneous B16-OVA tumor models	[169]
Platelets	High abundance in the blood; endothelium adhesion; tumor targeting and penetration; production of platelet secretory granules	Dox and cross-linked aPD-L1 nanogels	Postsurgical melanoma models	[170]
Mesenchymal stem cells	Tumor targeting; relatively abundant sources; immune regulation	Dox-loaded liposomes	Subcutaneous tumor and lung metastasis models	[171]
Bacteria	Tumor targeting; inducing macrophage polarization; colonization in tumor tissues	PLGA-R848 and PLGA-DOX	Orthotopic breast cancer models	[172]

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
