# Peer review of "When Natural Compounds Meet Nanotechnology: Nature-Inspired Nanomedicines for Cancer Immunotherapy"

_pharmaceutics, 2022, doi:10.3390/pharmaceutics14081589_

Round 1
Reviewer 1 Report
Immunotherapy represents a relatively new therapeutic approach in oncology, and is considered the new weapon for cancer treatment after surgery, chemotherapy and radiotherapy. The strategy is based on the fact that immunotherapy activates and strengthens the patient's immune system by prompting it to attack diseased cells. At the same time, in the therapeutic field, nanotechnologies open the way to new horizons of treatment. On the other hand, in recent years numerous studies have demonstrated the ability of compounds of natural origin to represent an enormous potential reservoir of solutions for human health.
This review provides an overview of the significant advances of natural compounds in cancer immunotherapy achieved with the use of different nanomaterials.
The authors examined an interesting topic on the potential immunoregulatory mechanisms of natural compounds at each stage of the cancer-immunity cycle, and the topic was adequately described.
The introduction provides sufficient background and the other sections include clearly presented and comprehensively analyzed data.
Although these are still experimental studies, which require clinical validation capable of establishing both tolerability and efficacy, the manuscript is well written, presented and discussed and understandable to a specialized readership.
The reported literature used to write the article is adequate and present in sufficient quantity.
No changes or additions are required for this manuscript.
Author Response
Point 1: No changes or additions are required for this manuscript.
Response 1: We are very grateful to the reviewer for the affirmation and appreciation of our work.

Reviewer 2 Report
1.The review is not innovative enough. Similar review has been published, eg: " Natural Product-based nanoformulations for cancer therapy: Opportunities and Challenges" and " Nanotechnology in Enhancing Immunotherapy for Cancer Treatment: Current Effects and Perspective."
2.The language needs to be polished by native speakers, as there are many grammatical errors (such as line15 and 17,357), etc.
3. There is no reference in the third from last line of Table1.
4. A good review should summarize other people work or research results and come up with new and meaningful own points views, rather than just simple summarizing.
Author Response
Point 1: The review is not innovative enough. Similar review has been published, eg: " Natural Product-based nanoformulations for cancer therapy: Opportunities and Challenges" and "Nanotechnology in Enhancing Immunotherapy for Cancer Treatment: Current Effects and Perspective.
Response 1: Thanks for your comment. The objective of reviews is to provide a diverse range of researchers active in the field with an easily accessible guide to up-to-date information and in-depth discussion. Compared with previous reviews concerning natural compounds or nanomaterials in cancer therapy, our paper shares the following novelties:
- Reviews of the same field generally have their own emphases, thus providing more detailed and meaningful information from different perspectives. Cancer immunotherapy, despite its long history, has blossomed into fruition only in recent years. Different from the previous reviews that focus broadly on the advances of multiple forms of cancer immunotherapy, our review innovatively emphasizes and highlights the immunoregulatory activities and involved mechanisms of natural compounds in each step of the cancer immunity cycle. Such a new but logical and meaningful perspective could facilitate a better understanding of the basic principles to amplify natural compound-mediated immune response against cancer and their advantages and limitations, and thus will help promote the development of novel strategies to circumvent their drawbacks and achieve optimal clinical efficacy. To promote the applications of nature compounds in cancer immunotherapy, nanotechnology has been employed. Hence, we also summarize various drug delivery strategies and representative applications of nature-inspired nanomedicines for cancer immunotherapy with the advantages and disadvantages discussed. Though some reviews about the natural compounds or nanotechnology for cancer therapy have been published, most of the works emphasized the function and design of nanomaterials to address issues such as insufficient accumulation in tumor issues or severe side-effects of compounds. Different from them, we especially focus on the various nanotechnology-based drug combinations for cancer immunotherapy to break through the defects of the single natural compound and thus amplify anti-tumor immune responses. Such a systematic summary will provide guidance on the exploration of novel nature-inspired nanomedicines and the establishment of suitable delivery platforms for better efficacy.
- In fact, there is still a lack of systematic and comprehensive summaries to cover the latest progress concerning natural compounds as immunomodulators and nature-derived nanomedicines for cancer immunotherapy. This is what our manuscript covers and differs from others. For example, the first review (“Natural Product-based nanoformulations for cancer therapy: Opportunities and Challenges”) you mentioned mainly focuses on the far-ranging antitumor potential of natural product-based nanoformulations, particularly in the advantages of nanotherapeutics to overcome the limitations of conventional drug delivery system, of which the content covers classification and physico-chemistry properties of natural compounds, different nanoformulation design, and varied anticancer mechanisms of important classes of natural compounds at a single platform. Though this published review is related to “natural compounds”, “nanotechnology”, and “cancer therapy”, there are few contents involved with the immunoregulatory activity of natural compounds or nature-derived nanomedicines against cancer. As for the second review (“Nanotechnology in Enhancing Immunotherapy for Cancer Treatment: Current Effects and Perspective”) you listed, it is mainly about nanomedicines with enhanced immunotherapy for cancer treatment, especially the rise of nanotechnology as a solution to promote drug accumulation in tumor tissues and alleviate side-effects of common immunomodulators, as well as nanoparticle-enhanced cancer immunotherapy strategies (including intensified delivery of tumor vaccines and immune adjuvants, immune checkpoint inhibitor vehicles, targeting capacity to tumor-draining lymph nodes and immune cells, triggered releasing and regulating specific tumor microenvironments, and adoptive cell therapy enhancement effects). And numerous similar reviews concerning nanomedicines for cancer immunotherapy have also been published (e.g., Biomaterials, 2017, 148, 16-30; Nat Rev Drug Discov, 2019, 18(3), 175-196; Acc Chem Res, 2019, 52(6), 1543-1554), which reflects that the reviews on the same topic can be accepted and published due to their timely updated content or different perspectives. Though these published reviews are related to “nanotechnology” and “cancer immunotherapy”, they also do not involve with the immunoregulatory activity of natural compounds, especially the important immunoregulatory mechanisms against cancer, resulting in that the researchers in the related fields (such as pharmaceutics, materials science, pharmacology, natural product chemistry, and Chinese pharmacy) cannot quickly and exhaustively consult useful information about natural compounds for cancer immunotherapy.
In conclusion, this timely and overall manuscript is innovative and meaningful. We believe that this review can benefit many researchers, and can also be an inspirational source to facilitate further development of delivery science and technology. Thank you again.
Point 2: The language needs to be polished by native speakers, as there are many grammatical errors (such as line15 and 17,357), etc.
Response 2: Thanks for your constructive suggestion. As you suggested, our manuscript has been undergone English language editing by Multidisciplinary Digital Publishing Institute (MDPI). All the changes have marked in the revised manuscript.
Point 3: There is no reference in the third from last line of Table1.
Response 3: Thank you for the important comment. We have supplemented the relevant reference (Ref. 111) in Table 1.
Point 3: A good review should summarize other people work or research results and come up with new and meaningful own points views, rather than just simple summarizing.
Response 4: Thank you for your valuable comment. In the original version, we summarized the immunoregulatory activities of natural compounds to initiate or re-implement the self-sustaining cancer-immunity cycle, analyzed their existing limitations, and proposed potential solutions in section 2. We also fully discussed the advantages and disadvantages of various nanotechnology-based delivery strategies by analyzing various application examples in section 3, thus providing guidance on the design of novel nature-inspired nanomedicines for cancer immunotherapy. In addition, we have added more new and meaningful own points of view and discussions in the revised manuscript as you suggested, especially our opinions on the potential research directions in the future and suggestions for certain applications in section 3 “Nature-inspired nanomedicine for cancer immunotherapy”, thus reflecting the research value of natural compounds for cancer immunotherapy in more depth and providing a basis for facilitating their far-reaching development towards clinical translation. And all revisions were marked up using the “Track Changes” function as required.

Reviewer 3 Report
A very exciting and interesting subject !! congratulations !! limitations of use of natural compounds are immediately described demonstrating that the authors are aware of the situation. For example, krestin (line 205) is well presented with the major interest of this compound and the difficulty to clearly demonstrate the clinical efficacy of the drug.
Line 7: I disagree with sentence « because of their definite curative effect and low side effects » this sentence disagree with the scientific information about their limitation of use in humans may I suggest being more cautious and just indicate that these compounds are highly active at the clinical level…
The review on natural compounds and their activities on various steps of anticancer response is very interesting and the level on scientific knowledge is impressive. The limitation of use is well summarized and recommendation to work on synergy of activity is repeated in the manuscript.
Table 1: the word « synergy » is a little bit exaggerated as mathematical demonstration of synergy is quite impossible, may I suggest a more cautious word such as « improved » or « increased »
Line 505 I don’t understand the origin of the sentence concerning metformin? can you precise the origin of this information?
Author Response
Point 1: A very exciting and interesting subject !! congratulations !!
Response 1: Thank you very much for your recognition of our work.
Point 2: Line 7: I disagree with sentence « because of their definite curative effect and low side effects » this sentence disagree with the scientific information about their limitation of use in humans may I suggest being more cautious and just indicate that these compounds are highly active at the clinical level…
Response 2: We appreciate the reviewer’s constructive comment. As suggested, we have revised the sentence (line 37-39) to “In the past 50 years, several typical natural compounds (e.g., ginsenoside Rg3, indirubin, artemisinin, and paclitaxel) have been proven to be highly active at the clinical level”.
Point 3: Table 1: the word « synergy » is a little bit exaggerated as mathematical demonstration of synergy is quite impossible, may I suggest a more cautious word such as « improved » or « increased »
Response 3: Thank you for your suggestion. As suggested, we have revised the “Synergistic effect” into the “Improved effect” in Table 1.
Point 4: Line 505 I don’t understand the origin of the sentence concerning metformin? can you precise the origin of this information?
Response 4: Thanks for your comment and we are sorry that we did not state this sentence clearly. Cancer stem cells (CSCs) are responsible for recurrence, metastasis, and local and systemic therapy resistance. Accumulating evidence has suggested that CSCs can create an immunosuppressive microenvironment, resulting in immune evasion (Nat Rev Cancer, 2021, 21(8), 526-536). Preclinically, metformin (MET), a natural compound derived from the legume Galega officinalis, has been reported to selectively suppress CSC growth in different types of tumors (e.g., breast cancer, pancreatic cancer, glioblastoma, and colon cancer) (Cancer Res, 2009, 69(19), 7507-11). In our manuscript (line 505-512), the researchers first demonstrated the antitumor effect of DOX-SAL NPs on 4T1 tumors and then introduced MET in the original preparation to eradicate cancer stem cells, thus further improving the treatment efficacy in an immunologically “cold” tumor. To provide precise information, we have revised the sentence into “Moreover, the existence of cancer stem cells (CSCs) involves the immunosuppressive tumor microenvironment, resulting in the propagation and metastasis of tumors. To eradicate CSCs and further improve treatment efficacy in an immunologically “cold tumor”, Li et al. also utilized metformin (MET) (a natural compound derived from the legume Galega officinalis) as a CSC-targeting agent and co-loaded it with DOX into SA-modified liposomes, of which the combination therapy with aPD-1 mAb showed increased benefit on both primary tumor inhibition and metastasis suppression.” and supplemented relevant reference (Ref. 117, 118, 119) for better understanding.
Reviewer 4 Report
The manuscript “When natural compounds meet nanotechnology: nature-inspired nanomedicines for cancer immunotherapy” is well written and organized.
In my opinion the review can be published
Author Response
Point 1: The manuscript “When natural compounds meet nanotechnology: nature-inspired nanomedicines for cancer immunotherapy” is well written and organized. In my opinion the review can be published.
Response 1: We are very grateful to the reviewer for the affirmation and appreciation of our work.

Round 2
Reviewer 2 Report
The revised version is suitable for acceptance.
This manuscript is a resubmission of an earlier submission. The following is a list of the peer review reports and author responses from that submission.